# FUS (fused in sarcoma) is a component of the cellular response to topoisomerase I–induced DNA breakage and transcriptional stress

Maria Isabel Martinez-Macias[1] (iD), Duncan AQ Moore[1], Ryan L Green[2], Fernando Gomez-Herreros[1,3] (iD), Marcel Naumann[4,6], Andreas Hermann[4,5,6], Philip Van Damme[7], Majid Hafezparast[2] (iD), Keith W Caldecott[1] (iD)

**FUS (fused in sarcoma) plays a key role in several steps of RNA metabolism, and dominant mutations in this protein are associated with neurodegenerative diseases. Here, we show that FUS is a component of the cellular response to topoisomerase I (TOP1)–induced DNA breakage; relocalising to the nucleolus in response to RNA polymerase II (Pol II) stalling at sites of TOP1-induced DNA breaks. This relocalisation is rapid and dynamic, reversing following the removal of TOP1-induced breaks and coinciding with the recovery of global transcription. Importantly, FUS relocalisation following TOP1-induced DNA breakage is associated with increased FUS binding at sites of RNA polymerase I transcription in ribosomal DNA and reduced FUS binding at sites of RNA Pol II transcription, suggesting that FUS relocates from sites of stalled RNA Pol II either to regulate pre-mRNA processing during transcriptional stress or to modulate ribosomal RNA biogenesis. Importantly, FUS-mutant patient fibroblasts are hypersensitive to TOP1-induced DNA breakage, highlighting the possible relevance of these findings to neurodegeneration.**

## Introduction

Amyotrophic lateral sclerosis (ALS) is a motor neuron disease with significant phenotypic variability but with some common pathological and genetic characteristics (reviewed in references 1, 2, 3). For example, mutation and/or toxic aggregation of RNA-binding proteins such as TAR DNA binding protein (TDP-43) and fused in sarcoma (FUS) have been associated with ALS (4, 5, 6, 7). In recent years, mutations in several additional RNA-binding proteins have been associated with neurodegenerative diseases, including EWS (EWSR1), TAF15 (8), hnRNPA1, hnRNP A2B1 (9), and ataxin-2 (10),

supporting the notion that defects in RNA metabolism can induce neurodegeneration (11, 12, 13).

ALS is the most common adult-onset motor neuron disease and is characterized by progressive degeneration of motor neurons. Although most cases of ALS are sporadic (sALS), 5–10% of cases have a familial history (fALS) (reviewed in references 2, 11, 14). It is thought that mutations in TDP-43 and FUS each account for 1–5% of fALS with a hexanucleotide repeat expansion in *C9ORF72* accounting for ~40% (2, 11, 14). FUS is a heterogeneous nuclear ribonucleoprotein (hnRNP) that belongs to the FET/TET family of RNA-binding proteins, including TAF15 and EWS (15, 16, 17, 18). FUS modulates multiple aspects of RNA metabolism, including transcription, splicing, microRNA processing, and mRNA transport (reviewed in references 18, 19, 20). Consequently, it has been proposed that ALS mutations cause pathological changes in FUS-regulated gene expression and RNA processing, due either to loss of normal FUS function, toxic gain of function, or both.

There is increasing evidence that FUS is also a component of the cellular response to DNA damage (21, 22, 23, 24). For example, FUS is phosphorylated by the DNA damage sensor protein kinases ATM and/or DNA-PK following treatment of cells with ionising radiation (IR) or etoposide (25, 26), and FUS deficiency in mice is associated with increased sensitivity to IR and elevated chromosome instability (27, 28). In addition, FUS accumulates at sites of laser-induced oxidative DNA damage in a manner that is dependent on the DNA strand break sensor protein, PARP1 (21,22). FUS interacts directly with poly (ADP-ribose), the RNA-like polymeric product of PARP1 activity, possibly promoting its concentration in liquid compartments and recruitment at DNA strand breaks (21, 22, 29). FUS reportedly also promotes the repair of DNA double-strand breaks (DSBs) by the nonhomologous end joining (NHEJ) and homologous recombination pathways for DSB repair (21, 23). Finally, FUS is present at sites of transcription at which RNA polymerase II (Pol II) is stalled by UV-induced DNA lesions and may

[1]Genome Damage and Stability Centre, School of Life Sciences, University of Sussex, Falmer, Brighton, England    [2]Neuroscience, School of Life Sciences, University of Sussex, Falmer, Brighton, England    [3]Instituto de Biomedicina de Sevilla, Hospital Virgen del Rocio-Centro Superior de Investigaciones Científicas-Universidad de Sevilla, Seville, Spain    [4]Department of Neurology, Technische Universität Dresden, and German Center for Neurodegenerative Diseases (DZNE), Dresden, Germany    [5]Center for Transdisciplinary Neurosciences Rostock, University Medical Center Rostock, University of Rostock, Rostock, Germany    [6]Translational Neurodegeneration Section "Albrecht-Kossel", Department of Neurology, University Medical Center Rostock, University of Rostock, Rostock, Germany    [7]University of Leuven, Leuven, Belgium

Correspondence: mm761@sussex.ac.uk; k.w.caldecott@sussex.ac.uk

    

facilitate the repair of R-loops or other nucleic acid structures induced by UV-induced transcription-associated DNA damage (24).

The observation that several other RNA-processing factors, in addition to FUS, are also implicated in the DNA damage response suggests that there is considerable cross-talk between these processes (30). However, the nature of the endogenous sources of DNA damage that might trigger a requirement for FUS and/or other RNA-processing factors is unknown. Of particular threat to neural maintenance and function is DNA damage induced by topoisomerases, a class of enzymes that remove torsional stress from DNA by creation of transient DNA strand breaks (31). Usually, these breaks are resealed by the topoisomerase enzyme at the end of each catalytic cycle, but on occasion, they can become abortive and require cellular DNA single- or DSB repair pathways for their removal. If not repaired rapidly or appropriately, topoisomerase-induced breaks can lead to chromosome translocations and genome instability in proliferating cells, and cytotoxicity and/or cellular dysfunction in post-mitotic cells. This is illustrated by the existence of hereditary neurodegenerative diseases in which affected individuals harbour mutations in tyrosyl DNA phosphodiesterase 1 (TDP1) or tyrosyl DNA phosphodiesterase 2 (TDP2) (32,33), DNA repair proteins with activities dedicated to removing trapped topoisomerases from DNA breaks (32, 33, 34).

To further address the relationship between ALS and endogenous DNA damage, we have examined the response of FUS to topoisomerase-induced DNA damage. Here, using a variety of different cell types, including human spinal motor neurons, we show that FUS is a component of the cellular response to transcriptional stress induced by topoisomerase I (TOP1)–associated DNA breakage. Importantly, we find that HeLa cells and ALS patient fibroblasts expressing mutant FUS are hypersensitive to TOP1-induced DNA breakage, highlighting the possible relevance of our findings to ALS disease pathology.

# Results

## Normal rates of DSB repair in FUS-mutated ALS patient fibroblasts

It has been reported that siRNA-mediated depletion of FUS results in reduced DNA DSB repair and that FUS is involved in this process (21, 23). To examine whether this observation is relevant to topoisomerase-induced DNA breaks in the context of ALS, we compared fibroblasts from an ALS patient harbouring the common disease mutation FUS$^{R521H}$ and an unaffected sibling for DSB repair kinetics following treatment with the genotoxins camptothecin (CPT) or etoposide. These genotoxins exert their cytotoxic effects by triggering the abortive activity of TOP1 and topoisomerase II (TOP2), thereby inducing TOP1- and TOP2-induced DNA breaks, respectively. Both TOP1- and TOP2-induced DNA breaks are physiologically relevant sources of DNA damage that are implicated in neurodegeneration (32, 33, 35). Surprisingly, immunostaining for γH2AX, an indirect marker of DSB sites, failed to reveal any difference in the kinetics of DSB induction and removal between unaffected and ALS fibroblasts following treatment with either CPT or etoposide (Fig 1A and B). We also failed to observe any difference in kinetics of DSB induction and repair following treatment with IR, which induces DSBs independently of topoisomerase activity (Fig 1C). Together, these experiments suggest that ALS pathology arising from FUS mutation does not result from defects in the repair of DSBs induced by topoisomerases or oxidative stress.

## FUS relocalises to the nucleolus in response to TOP1-induced DNA breakage

Despite the results described above, we considered it likely that FUS is part of the cellular response to DNA damage because this protein relocalises to sites of laser-induced DNA damage (21, 22). We, therefore, employed a HeLa cell line stably expressing GFP-FUS (36) to examine the subcellular localization of this protein before and after treatment with CPT, a more physiologically relevant source of DNA damage treatment. Strikingly, we observed a rapid relocalisation of GFP-FUS into nuclear foci (Fig 2A). This was not the case following IR, however, suggesting that FUS relocalisation was not due simply to the induction of DSBs. Subsequent experiments, in which we co-stained cells for fibrillarin, revealed that the sites of GFP-FUS accumulation following CPT treatment were restricted to nucleoli and/or nucleolar caps (Fig 2B). This relocalisation was not an artefact of FUS overexpression or the GFP tag because a similar phenomenon was observed with endogenous FUS in A549 cells (Fig 2C) and in primary human fibroblasts (Fig 2D). Notably, we observed a similar response for GFP-tagged TDP-43, another ALS-associated protein involved in RNA metabolism, albeit to a lesser extent (Fig S1B). A similar response was observed for ALS-associated mutant derivatives of both GFP-FUS (GFP-FUS$^{R521C}$ and GFP-FUS$^{P525L}$; Fig S1A) and GFP-TDP-43 (GFP-TDP-43$^{G298S}$ and GFP-TDP-43$^{M337V}$; Fig S1C), suggesting that ALS is not caused simply by a defect in the relocalisation of these proteins to nucleoli.

Because ALS is characterized by progressive neurodegeneration (14), we wished to confirm that the relocalisation of FUS to nucleoli induced by CPT in cultured cell lines also occurred in neurons. Indeed, endogenous FUS similarly relocalised to nucleoli in mouse cortical neurons following CPT treatment (Fig 3A). Similarly, FUS-GFP relocalised to nuceoli following CPT treatment in patient-derived human-induced pluripotent stem cells (hiPSCs) that were gene edited to express this protein from the endogenous FUS locus, following their differentiation into human spinal motor neurons (Fig 3B) (37). That we successfully generated motor neurons in these experiments was confirmed by immunostaining for MAP2 and ChAT (Fig 3B and C) and by the absence of detectable expression of SOX2 (data not shown). Interestingly, FUS$^{P525L}$-GFP similarly relocalised to nucleoli/nucleolar caps in isogenic human spinal motor neurons following CPT treatment (Fig S2), again suggesting that FUS relocalisation is not markedly affected by ALS-associated mutations.

## FUS relocalisation following TOP1-induced DNA breakage is triggered by RNA Pol II inhibition

Both CPT and IR induce DNA strand breaks, but only CPT induced the relocalisation of FUS to nucleoli. Consequently, we considered it unlikely that the nucleolar relocalisation was triggered directly by DNA breaks. Consistent with this idea, although the SSB sensor

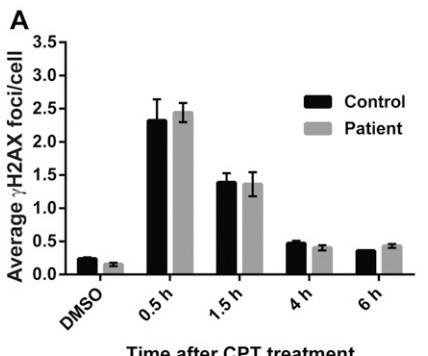

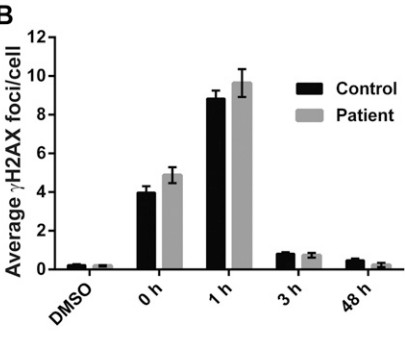

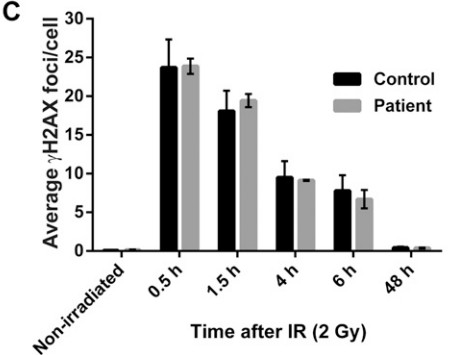

**Figure 1. Normal DSB repair kinetics in fibroblasts from FUS$^{R521H}$ ALS fibroblasts.**
**(A–C)** DSBs were quantified as γH2AX foci in G1-arrested ALS-FUS$^{R521H}$ patient and unaffected sibling (Control) fibroblasts. Cells were DMSO-treated, non-irradiated, or treated as indicated with either (A) 4 $\mu$M CPT for 45 min, (B) 20 $\mu$M etoposide for 30 min, or (C) 2 Gy IR. γH2AX foci were counted in CENPF-negative cells at the indicated time points after treatment. Data are the average (±SEM) of three independent experiments.

protein PARP1 triggers GFP-FUS recruitment to sites of oxidative DNA damage (21, 22), inhibitors of PARP1 failed to disrupt the relocalisation of FUS to nucleoli/nucleolar caps following CPT treatment (Fig 4A). Similarly, inhibitors of the ataxia telangiectasia and Rad3 related (ATR), ataxia telangiectasia mutated (ATM), and DNA-PK protein kinases that are the primary sensors of DSBs also failed to prevent FUS movement to nucleoli (Fig 4B). Importantly, immunostaining for 53BP1 confirmed that whereas only CPT triggered this FUS relocalisation, DSBs were induced both by CPT and IR (Fig 4C).

Because FUS is implicated in regulating gene transcription (38) and because TOP1-induced DNA breaks are potent inhibitors of RNA polymerase progression (39, 40), we considered the possibility that the relocalisation of FUS to nucleoli following CPT treatment was triggered by RNA polymerase blockage. Consistent with this idea, treatment with CPT reduced the rate of transcription in the nucleus and nucleolus, as measured by 5-ethynyl uridine (EU) pulse labelling (Fig 5A, left). Moreover, the relocalisation of FUS to the nucleolus was rapidly reversible, with the exit of this protein from the nucleolus following CPT removal coincident with the recovery of global transcription (Fig 5B). This was not the case for IR, which consistent with its inability to induce FUS relocalisation did not reduce the global rate of transcription (Fig 5A, right). To further examine whether FUS relocalisation was triggered by transcriptional stress, we used the RNA polymerase inhibitor actinomycin D. Indeed, incubation with actinomycin D at concentrations (4 $\mu$M) that inhibit both RNA polymerase I (Pol I) and RNA Pol II induced rapid FUS relocalisation to the nucleolus (Fig 5C). However, incubation with lower concentrations of actinomycin D (5 nM) that inhibit only Pol I did not trigger FUS relocalisation, suggesting that it is the

inhibition of Pol II that triggers FUS accumulation at nucleoli (Fig 5C). Consistent with this idea, incubation with CX5461 (41), a specific inhibitor of Pol I, failed to trigger FUS relocalisation (Fig 5C). Moreover, CX5461 strongly suppressed CPT-induced relocalisation of FUS to nucleoli, with only 25% of the cells showing residual nucleolar FUS (Fig 5C). Because CX5461 prevents the initiation of transcription by Pol I, this result suggests that FUS relocalisation following TOP1-induced DNA breakage requires Pol I to be engaged in transcription. Interestingly, 5,6-dichloro-1-$\beta$-D-ribo-furanosylbenzimidazole (DRB), another RNA synthesis inhibitor, also triggered the recruitment of wild-type GFP-FUS (and the ALS-associated mutants GFP-FUS$^{R521C}$ and GFP-FUS$^{P525L}$) to nucleoli (Figs 5D and S3), and this response was also impaired by CX5461 treatment (Fig 5D). Collectively, these data suggest that the accumulation of FUS at nucleoli requires ongoing Pol I transcriptional activity and is triggered by Pol II inhibition.

### Chromatin binding by FUS is increased at transcriptionally active rDNA following TOP1-induced DNA breakage

To examine whether the relocalisation of FUS observed by immunofluorescence reflects a change in FUS activity at the molecular level, we used chromatin immunoprecipitation to compare the level of FUS binding at the ribosomal DNA (rDNA) locus and at *FOS*, a Pol II transcribed gene and target for TOP1-induced DNA breakage (42, 43). Indeed, these experiments revealed that FUS was bound to the transcribed region of *FOS* following induction of the latter gene by the calcium ionophore A23187, and that this binding decreased following CPT treatment (Fig 6A). We noted that the total level of *FOS*

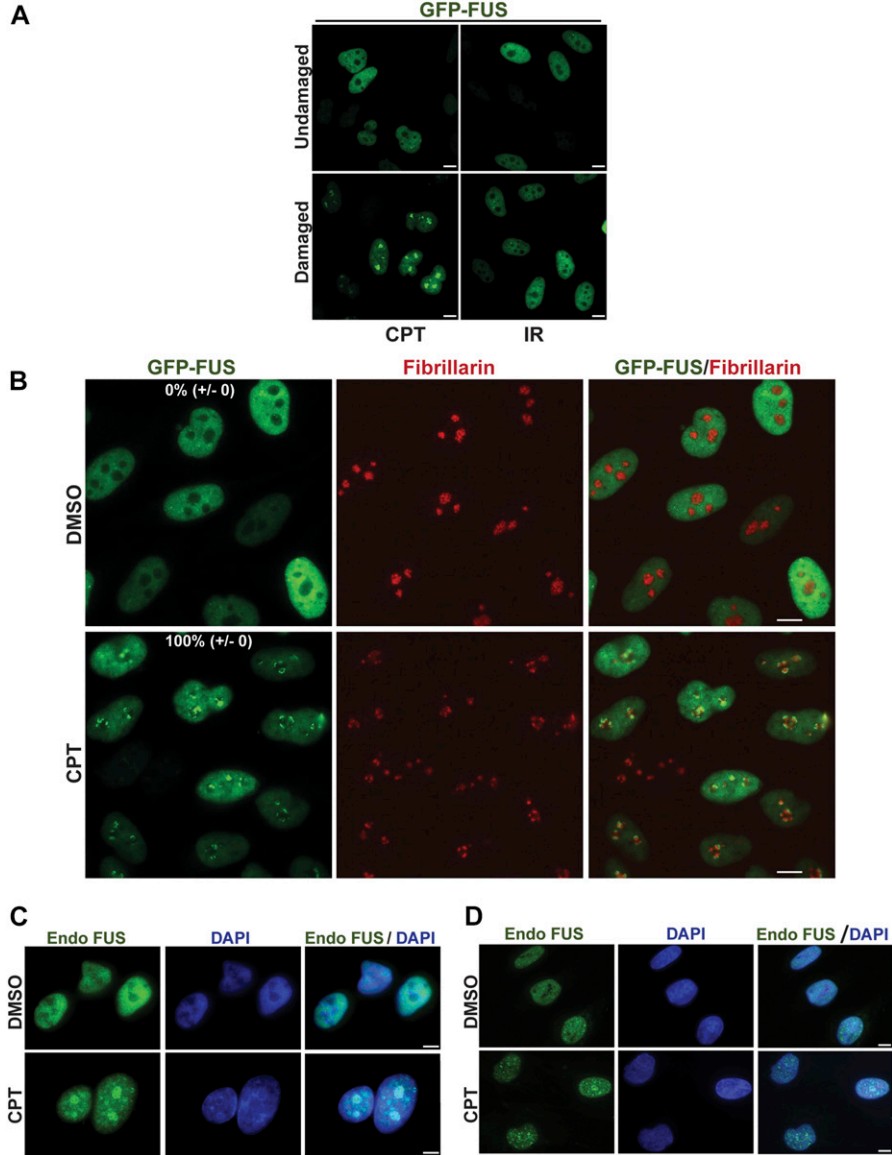

**Figure 2.   FUS relocalises to nucleoli in response to TOP1-induced DNA breakage.**
**(A)** GFP-FUS fluorescence was detected in untreated HeLa cells and in HeLa cells following treatment with the TOP1 poison CPT (4 μm) for 45 min or 30 min after IR (2 Gy). Scale bars, 10 μm. **(B)** GFP-FUS fluorescence was detected in HeLa cells treated with DMSO vehicle or with 4 μm of CPT for 1 h. Anti-fibrillarin immunostaining was used to stain nucleoli. Numbers are the mean percentage (±SD) of GFP-positive cells with GFP-FUS nucleolar localization in three independent experiments (50 cells per experiment). Scale bars, 10 μm. **(C, D)** Endogenous FUS was detected by indirect immunofluorescence in A549 cells (C) and primary human fibroblasts (D) treated with DMSO vehicle or 4 μM CPT for 45 min. Scale bars, 5 μm.

mRNA was reduced following CPT treatment, and that the relative amount of spliced mRNA was increased, confirming that *FOS* expression was impacted by TOP1-induced DNA breakage in these experiments (Fig 6B and C). In contrast, consistent with our immunofluorescence experiments, FUS binding at the rDNA locus increased >fourfold following treatment with CPT (Fig 6D). Notably, CPT increased FUS binding at the 18S and ITS-1 regions of rDNA but not at the promoter, again suggesting that FUS relocalises to the region of rDNA genes that are engaged by Pol I. Indeed, in agreement with this idea and with our immunofluorescence experiments, co-incubation with the Pol I inhibitor CX5461 greatly reduced FUS recruitment at rDNA in response to CPT (Fig 6D). Because CPT also inhibited nucleolar transcription (see Fig 5A), we considered it likely that FUS relocalised to rDNA genes at which Pol I was engaged but stalled. Consistent with this idea, chromatin immunoprecipitation experiments using the antibody S9.6 identified elevated levels of RNA/DNA hybrid and/or R-loop structures at the 18S and ITS1 regions

of rDNA, following CPT treatment (Fig 6E). Similar to FUS relocalisation and binding, the appearance of these RNA/DNA hybrids at the rDNA was dependent on Pol I activity (Fig 6E).

## ALS patient fibroblasts and HeLa cells expressing ALS-associated FUS mutations are hypersensitive to TOP1-induced DNA breakage

Collectively, our results suggest that FUS is part of a cellular response to Pol II transcriptional stress, including that triggered by TOP1-induced DNA breaks, in which this RNA-processing factor relocalises to sites of ribosomal RNA (rRNA) transcription. To determine whether this response is important for cell survival, we compared the sensitivity of fibroblasts from a patient with ALS harbouring the mutation $FUS^{R521H}$ with those of the unaffected sibling control. Notably, the ALS patient fibroblasts exhibited increased sensitivity to CPT (Fig 7A, left). In contrast, these cells exhibited normal levels of sensitivity to IR, consistent with the

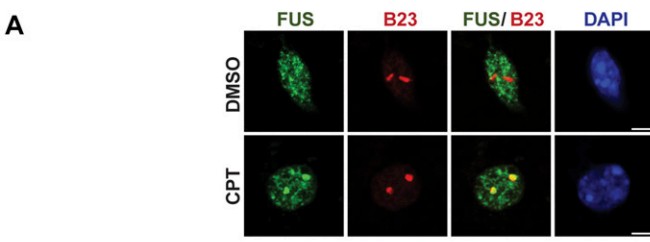

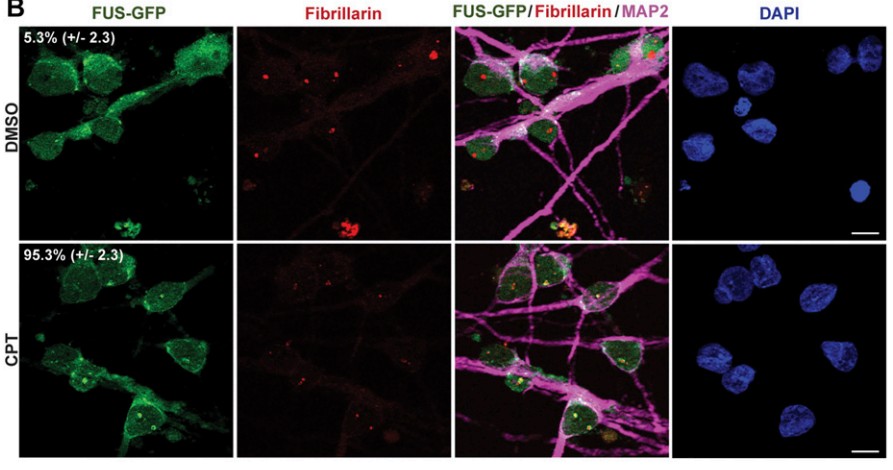

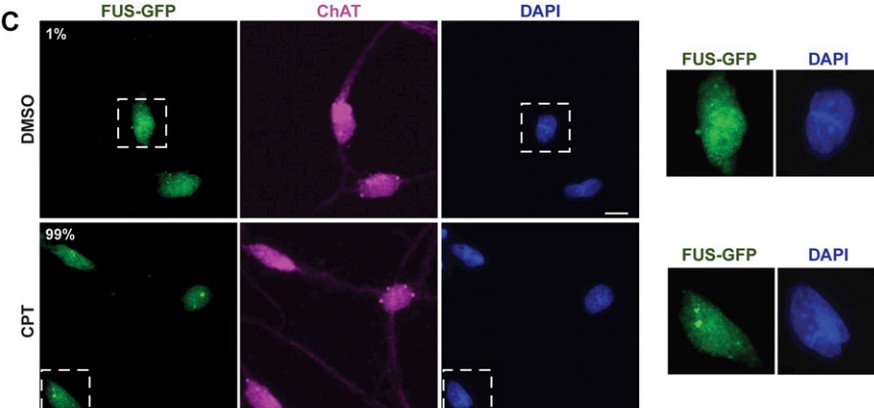

**Figure 3. FUS relocalises to nucleoli in response to TOP1-induced DNA breakage in neurons.**
**(A)** Endogenous FUS and B23 (nucleophosmin; nucleolar marker) were detected by indirect immunofluorescence in mouse cortical neurons following incubation with DMSO vehicle or 4 $\mu$M CPT for 45 min. Scale bar, 5 $\mu$m. **(B)** FUS-GFP was detected by direct fluorescence in human spinal motor neurons after treatment with either DMSO vehicle or 4 $\mu$M CPT for 1 h. Fibrillarin and MAP2 were detected by indirect immunofluorescence to stain nucleoli and neurons, respectively. Numbers are the mean percentage (±SD) of MAP-positive cells with FUS-GFP nucleolar localization in three independent experiments (50 cells per experiment). Scale bar, 10 $\mu$m. **(C)** FUS-GFP was detected by direct fluorescence in human spinal motor neurons after treatment with either DMSO vehicle or 4 $\mu$M CPT for 1 h. Choline acetyltransferase (ChAT) was detected by indirect immunofluorescence to confirm the identity of motor neurons. Numbers are the percentage of ChAT-positive cells with FUS-GFP nucleolar localization (n = 100 cells). Scale bar, 10 $\mu$m. Zoomed areas are shown (right).

relative lack of measurable impact of IR on transcription (Fig 7A, right). To rule out that the increased sensitivity to CPT was due to differences in genetic background between the unaffected and ALS fibroblasts, we compared HeLa cells expressing either wild-type GFP-FUS or the ALS-associated mutant derivatives GFP-FUS[R521C] and GFP-FUS[P525L] (Fig 7B). Similar to ALS patient fibroblasts, HeLa cells expressing GFP-FUS[R521C] and GFP-FUS[P525L] were more sensitive to CPT than HeLa cells expressing wild-type GFP-FUS, supporting the notion that FUS mutations associated with ALS confer hypersensitivity to TOP1-induced DNA breakage.

## Discussion

FUS is a hnRNP that belongs to the FET/TET family of RNA-binding proteins comprising FUS/TLS, EWSR1, and TAF15 (15,18). Under physiological conditions, FUS is localized primarily in the nucleus, although it can shuttle between the nucleus and cytoplasm (44). FUS binds a number of nucleic acids and nucleic acid–like structures including single- and double-stranded DNA, RNA, and poly (ADP-ribose), but the precise biochemical function of FUS is unclear (15, 18, 21, 22). Of these activities, RNA binding is most likely critical because FUS influences the synthesis and processing of a large number of pre-mRNAs (45). Consistent with this, FUS interacts with a variety of transcription factors and with the C-terminal domain of RNA Pol II, regulating its phosphorylation during transcription (38, 46). Notably, ALS-associated mutations in FUS have been reported to reduce binding to Pol II (47) and transcriptionally active chromatin (48).

Recently, it was suggested that FUS is required for the repair of DNA DSBs (21, 23). RNAi-mediated depletion of FUS and FUS carrying familial ALS mutations was reported to reduce the efficiency of DSB

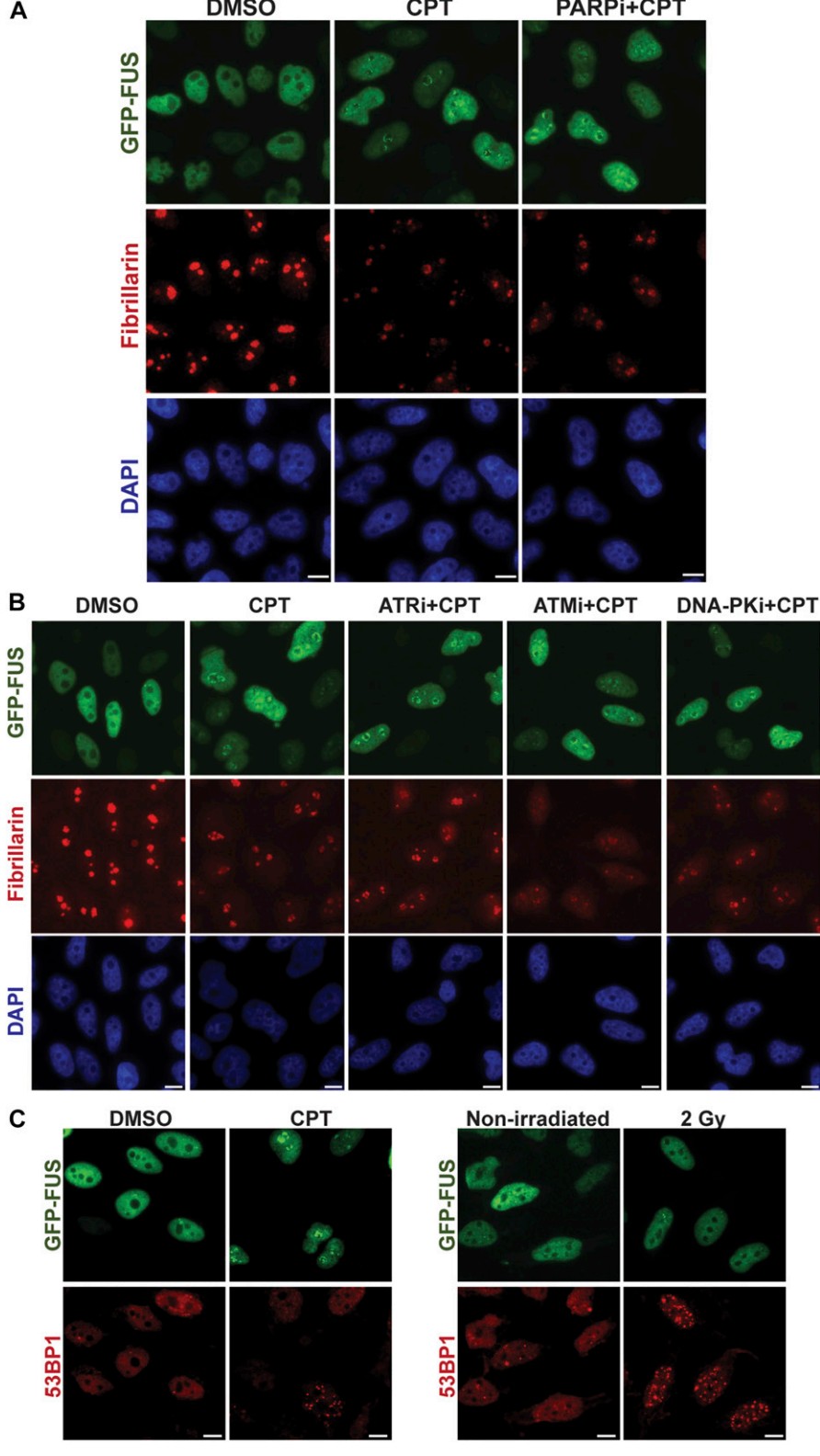

**Figure 4. FUS nucleolar relocalisation is not a direct response to DNA breakage.**
**(A)** GFP-FUS fluorescence and fibrillarin immunofluorescence (a nucleolar marker) were detected in HeLa cells preincubated or not with PARP inhibitor (10 μM KU0058948; PARPi) for 1 h before a 1-h incubation with 4 μM CPT. The cells were counterstained with DAPI to detect nuclei. Scale bars, 10 μm. **(B)** HeLa cells were preincubated or not with 5 μM ATR kinase inhibitor (ATRi) for 15 min, 10 μM ATM inhibitor (ATMi, KU55933) for 30 min, or 5 μM DNA-PK inhibitor (DNA-PKi, NU7441) for 1 h before a 1-h incubation with 4 μM CPT. The cells were counterstained with DAPI to detect nuclei. Scale bars, 10 μm. **(C)** GFP-FUS fluorescence and 53BP1 immunofluorescence (a DSB marker) were detected in HeLa cells following treatment with 4 μM of CPT for 1 h (left) or following IR (2 Gy) (right). Scale bars, 10 μm.

repair by both NHEJ and homologous recombination, the two primary mechanisms by which DSBs are repaired. However, we failed to observe a defect in our experiments in DSB repair in

ALS fibroblasts harbouring the pathogenic FUS mutation, R521H, following either IR or following treatment with etoposide or CPT. A similar lack of defect in repair rate was reported in human

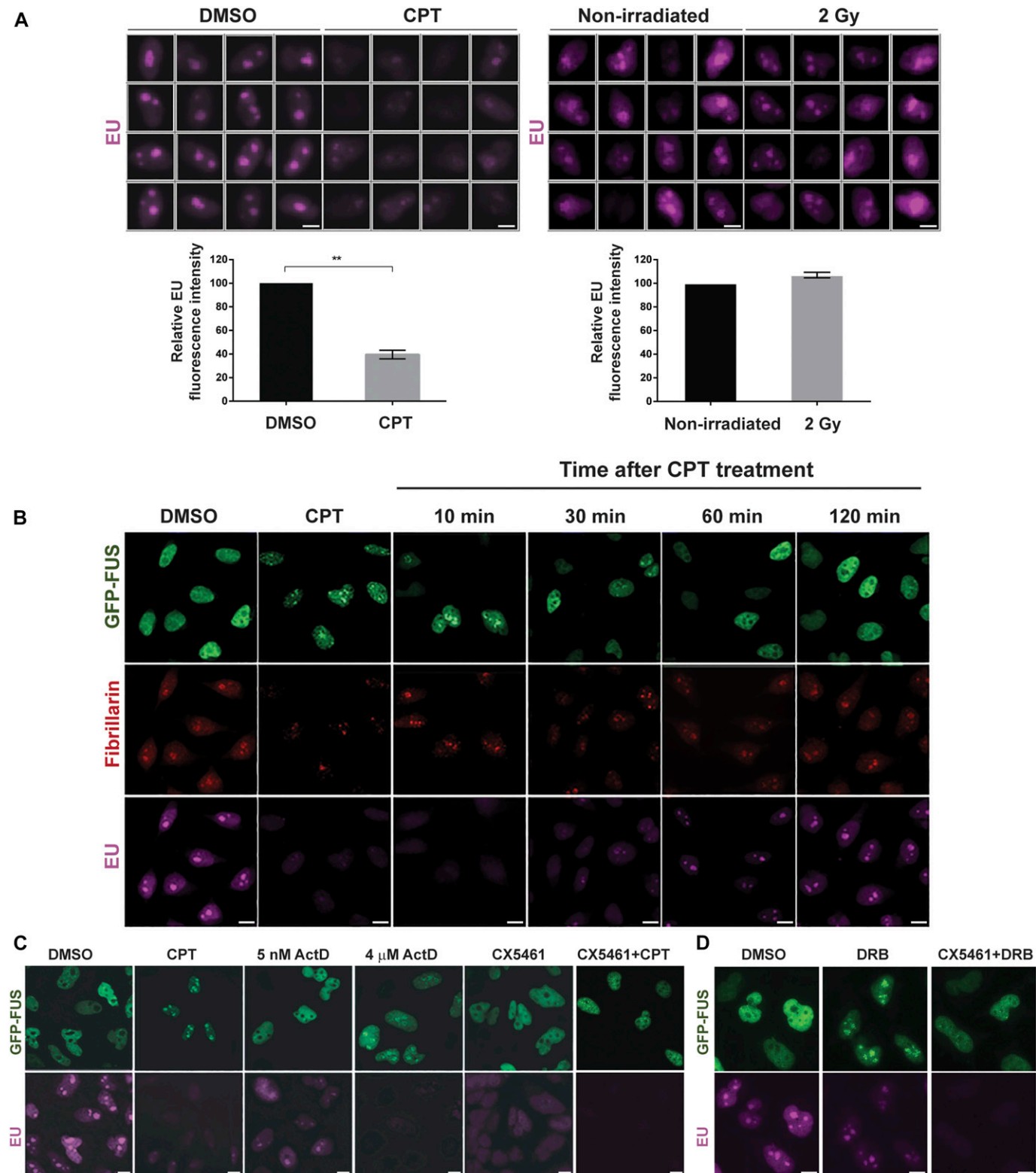

**Figure 5.  FUS nucleolar relocalisation is reversible and triggered by transcriptional stress.**
**(A)** Top: representative ScanR images of 5-EU pulse labelling in HeLa cells treated with DMSO vehicle or 4 μM CPT for 1 h (left) or following irradiation (2 Gy) (right). Bottom: quantification of EU signal from >500 cells per sample using Olympus ScanR analysis software. Data are the mean (±SEM) of three independent experiments. Statistically significant differences are indicated (two-tailed t test; **$P < 0.01$). **(B)** GFP-FUS, fibrillarin, and EU were detected by direct fluorescence (GFP), immunofluorescence (fibrillarin), or Click Chemistry (EU) in HeLa cells treated with DMSO vehicle or 4 μM CPT for 1 h (CPT), and in CPT-treated HeLa cells following subsequent incubation in

patient-derived motor neurons (37). Notably, we have also failed to detect a difference between ALS fibroblasts and unaffected sibling control in the rate of DNA single-strand break (SSB) repair, following treatment with $H_2O_2$ or CPT (unpublished observations). This contrasts with a recent study in which FUS was reported to promote SSB repair by facilitating XRCC1 recruitment and ligation at SSBs (49). These conflicting results may reflect differences in the assays or cell lines used in the studies.

Although we did not detect a requirement for FUS for DSB repair, our current data support the idea that FUS is involved in the response to transcriptional stress arising from TOP1-induced DNA breakage. We discovered that FUS relocates from the nucleoplasm to the nucleolus in a rapid and dynamic manner following the TOP1-induced DNA breakage by CPT. TOP1 creates transient SSBs to release torsional constraints in DNA during processes such as DNA replication and transcription (31). Normally, these breaks are resealed by the topoisomerase at the end of each catalytic cycle, but on occasion they can become abortive and require cellular SSB or DSB repair pathways for their removal (reviewed in reference 31). Topoisomerase "poisons" such as CPT induce this type of DNA breakage by inhibiting the religation activity of TOP1 (reviewed in reference 50). The threat posed by topoisomerase-induced breaks is illustrated by the existence of human hereditary diseases in which the repair of these breaks is attenuated, resulting in neurodevelopmental pathology and/or progressive neurodegeneration (32, 33, 35, 51, 52). Importantly, the relocalisation of FUS to nucleoli triggered by TOP1-induced breakage raises the possibility that these lesions may also be an etiological factor in ALS, a possibility supported by our observations that this response to CPT was also detected in human iPSC-derived spinal motor neurons, and by the sensitivity of ALS patient-derived fibroblasts and HeLa cells expressing ALS-associated FUS mutations to CPT.

Under normal conditions, FUS was primarily detected outside of nucleoli, but accumulated in the nucleoli and/or at the nucleolar periphery in a rapid and dynamic manner following TOP1-induced DNA breakage. The relocalisation of FUS to the nucleolus was not a result of the DNA breaks per se because it was not induced by IR at doses that induced similar if not more DNA breaks than CPT. Rather, the relocalisation of FUS to nucleoli appeared to result from the Pol II transcriptional stress imposed by TOP1-induced breaks. Consistent with this conclusion, other inhibitors of Pol II such as actinomycin D and DRB similarly triggered FUS relocalisation to nucleoli (this work and [53, 54, 55]). Intriguingly, TOP1 is enriched in nucleoli and is associated with ribosomal genes, possibly to facilitate rRNA transcription (56,57). Curiously, however, treatment with DRB or CPT results in the relocalisation of TOP1 out of the nucleolus, a phenomenon that is opposite to the behaviour of FUS reported here (56). Whether or not these observations are mechanistically related remains to be determined, but it seems likely that they are part of the same physiological response to transcriptional stress.

We do not yet know what facilitates the movement of FUS to sites of rRNA synthesis, but unlike FUS recruitment at sites of oxidative DNA damage (21, 22), it does not appear to be regulated by PARP1 or by DNA damage response protein kinases and seems not to be affected by ALS causing FUS mutations. Importantly, the subcellular relocalisation of FUS to the nucleolus was paralleled at the molecular level because FUS binding at the rDNA locus increased following CPT treatment, as measured by chromatin immunoprecipitation experiments, and decreased at the transcriptionally active FOS locus; a gene at which TOP1-induced SSBs are known to be induced and inhibit Pol II progression (42, 43). Although the FUS relocalisation to nucleoli observed in our work appeared to be triggered by Pol II transcriptional stress, it required the rDNA loci at which FUS was recruited to be transcriptionally active. This was suggested by the recruitment of FUS specifically to regions of the rRNA gene body, but not the promoter, and by the decrease of FUS relocalisation to rDNA if the initiation of transcription by Pol I was inhibited by incubation with CX5461. The accumulation of FUS in nucleoli and/or at the periphery nucleolar may, thus, involve a direct, or indirect, association with incomplete or partially processed rRNA.

What is the purpose of FUS relocalisation to the nucleoli following Pol II inhibition? One possibility is that FUS protein relocalises to transcriptionally active rDNA loci to protect these sites from the pathological effects of TOP1-induced SSBs. For example, the transcriptional blockage induced by TOP1-induced breaks can promote the formation of potentially pathogenic R loops, and FUS has been reported to participate in the prevention and/or repair of R loop–associated DNA damage (24). Alternatively, perhaps FUS regulates the level of pre-rRNA synthesis and/or processing under conditions of transcriptional stress. Finally, because FUS is also strongly implicated in pre-mRNA splicing/processing (reviewed in references 12, 13, 19), perhaps the relocalisation of FUS to nucleoli helps repress aberrant pre-mRNA processing following Pol II stalling (24, 38). Although one or more of the roles played by FUS following relocalisation may be defective in ALS, it should be noted that the relocalisation process itself is not defective because this phenomenon was similarly observed by FUS (and TDP-43) proteins that harbour ALS-associated mutations.

In summary, we show here that FUS (and TDP-43) relocalises from the nucleoplasm to sites of nucleolar rRNA synthesis in response to Pol II transcriptional stress, including that induced by abortive TOP1 DNA breakage. To our knowledge, these data are the first to demonstrate that FUS is mobilised in response to topoisomerase-induced DNA breaks. We propose that FUS moves from sites of stalled Pol II to sites of Pol I activity either to regulate pre-mRNA synthesis and/or processing during transcriptional stress, or to modulate some as yet unidentified aspect of rRNA biogenesis. Consequently, we suggest that TOP1-induced DNA breakage is a possible etiological factor in ALS pathology.

CPT-free medium for 10, 30, 60, or 120 min. All cells were pulse-labelled with EU for 30 min before fixation. Scale bars, 10 μm. **(C)** GFP-FUS and EU were detected in HeLa cells treated with either DMSO vehicle, 4 μM CPT for 1 h, actinomycin D (5 nM or 4 μM) for 1 h or 10 μM CX5461 for 3 h. Where indicated, the cells were preincubated with CX5461 for 3 h before a 1-h incubation with 4 μM CPT (CX5461+CPT). Scale bars, 10 μm. **(D)** GFP-FUS and EU were detected in HeLa cells treated with either DMSO or 100 μM DRB for 30 min. Where indicated, the cells were preincubated with CX5461 for 3 h before a 30-min incubation with 100 μM DRB (CX5461+DRB). Scale bars: (A): 5 μm; (B–D): 10 μm.

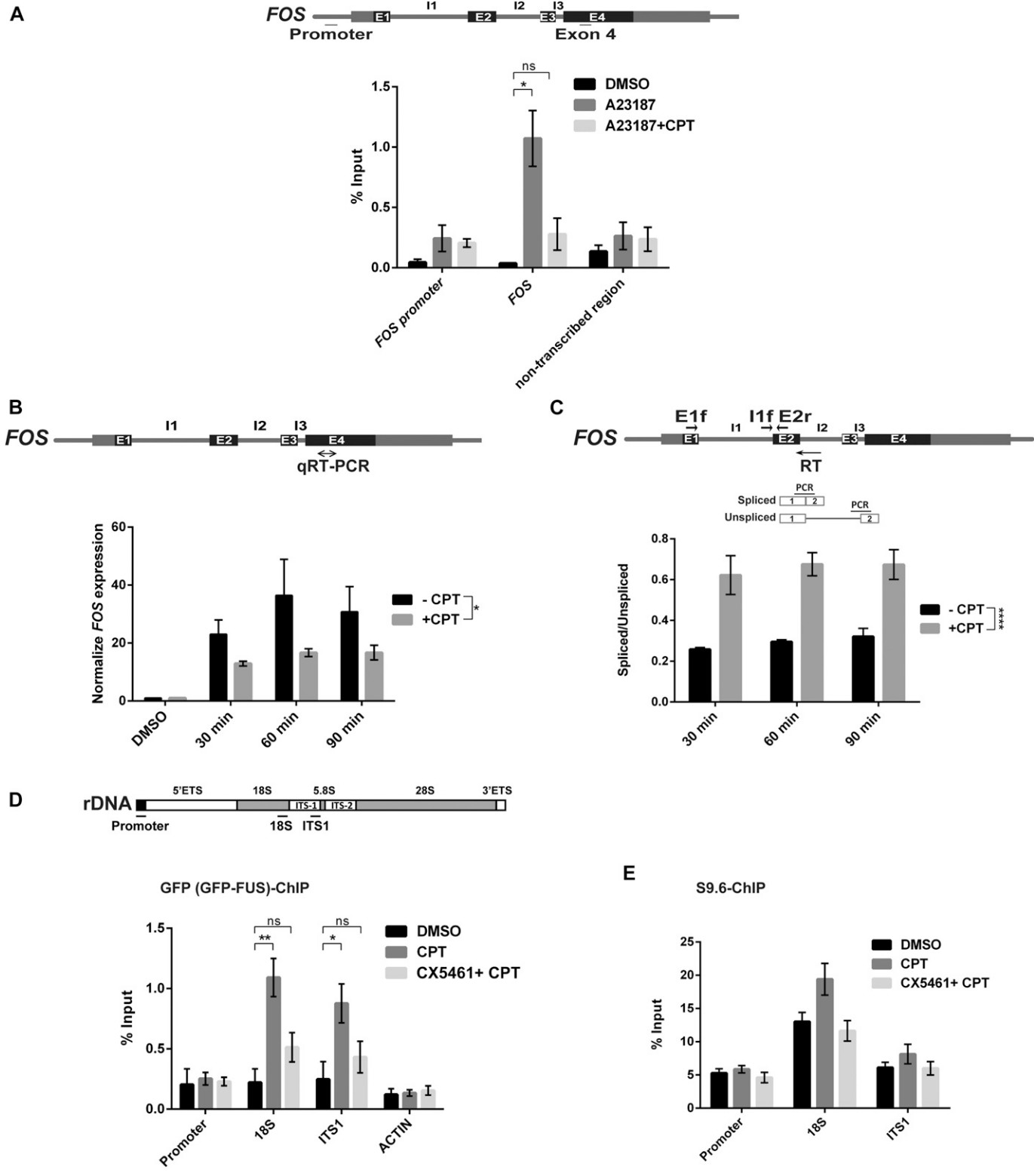

**Figure 6. FUS chromatin binding is increased at rDNA following TOP1-induced DNA breakage.**
**(A)** GFP-FUS binding at the indicated regions of *FOS* was quantified by chromatin immunoprecipitated/qPCR in HeLa cells stimulated with calcium ionophore (5 µM of A23187) for 30 min to induce *FOS* expression, in the presence or absence of 10 µM of CPT. Top: schematic showing the regions of the *FOS* promoter and exon 4 amplified by qPCR. A non-transcribed region of chromosome 5 was also amplified and quantified as a control. Data are the mean (±SEM) of four independent experiments. Statistically nonsignificant ("ns") and significant (two-tailed *t* test; *P < 0.05) differences are indicated. **(B)** Quantification of *FOS* mRNA in serum-starved HeLa cells stimulated with A23187 to induce *FOS* for the indicated times in the presence of DMSO vehicle (-CPT) or 10 µM CPT. Top: schematic of the *FOS* gene showing the region amplified by qRT-PCR.

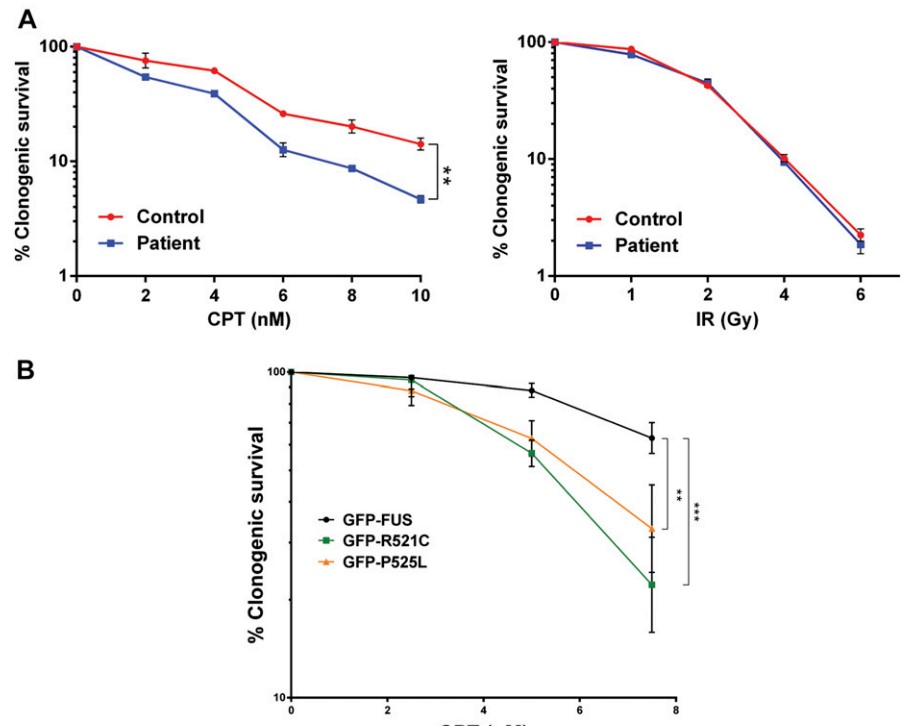

**Figure 7. ALS-associated FUS mutations confer hypersensitivity to TOP1-induced DNA breakage.**
**(A)** Clonogenic survival of fibroblasts from an ALS patient expressing FUS[R521H] and an unaffected sibling control (Control) in the presence of the indicated concentrations of CPT (left) or following irradiation at indicated doses (right). Data are the mean (±SEM) of three independent experiments. **(B)** Clonogenic survival of HeLa cells expressing wild-type GFP-FUS or the ALS-associated mutants GFP-FUS[R521C] (GFP-R521C) or GFP-FUS[P525L] (GFP-P525L) in the presence of the indicated concentrations of CPT. Data are the mean (±SEM) of four independent experiments. Statistically significant differences are indicated (two-factor ANOVA; **$P < 0.01$, ***$P < 0.001$).

# Materials and Methods

## Cell culture and drug treatments

The isogenic HeLa stable cell lines expressing inducible GFP-FUS, GFP-FUS mutants (GFP-FUS[R521C] and GFP-FUS[P525L]), GFP-TDP-43, and GFP-TDP-43 mutants (GFP-TDP-43[G298S] and GFP-TDP-43[M337V]) transgenes were kindly provided by Don Cleveland (University of California, US) and previously described in references 36, 57. These cells were grown in DMEM supplemented with 10% FCS, 2 mM glutamine, and the antibiotics penicillin (100 units/ml) and streptomycin (100 $\mu$g/ml). Transgene expression was induced with 4 $\mu$g/ml tetracycline for 48 h. Human A549 cells were grown in DMEM supplemented with 15% FCS and with penicillin/streptomycin and glutamine as above. hTERT-immortalised ALS fibroblasts derived from a 33-yr-old female ALS patient harbouring the heterozygous mutation R521H and a 44-yr-old clinically unaffected male sibling were grown in MEM containing 15% FCS, penicillin/streptomycin, and glutamine. All cells were grown at 37°C and 5% $CO_2$. Where indicated, the cells were treated with 2 Gy IR or with the following chemicals: CPT (Sigma-Aldrich) was employed at 4 $\mu$M or 10 $\mu$M for

the indicated time points, etoposide (VP16; Sigma-Aldrich) was used at 20 $\mu$M for 30 min, PARP inhibitor, KU0058948 hydrochloride (Axon), at 10 $\mu$M for 1 h, ATR inhibitor, ATR kinase inhibitor II (Calbiochem), at 5 $\mu$M for 15 min, ATM inhibitor, KU55933 (Abcam), at 10 $\mu$M for 30 min, DNA-PK inhibitor, NU7441 (Tocris), at 5 $\mu$M for 1 h, actinomycin D (Sigma-Aldrich) at 5 nM for 1 h for inhibition of RNA Pol I, and 4 $\mu$M for inhibition of both RNA Pol I and II. The RNA Pol I inhibitor, CX-5461 (Medchem Express), was used at 10 $\mu$M for 3 h. DRB was used at 100 $\mu$M for 30 min. Calcium ionophore for *FOS* expression, A23187 (Sigma-Aldrich), was used at 5 $\mu$M for the time indicated in the text.

## Mouse cortical neurons

CD1 embryonic mouse brains (18 days post coitum) were removed from the cranial cavity and placed in HBSS (10594243; Invitrogen) containing 1% P/S (11528876; Gibco). The meninges were removed, the hemispheres separated, and coronal slices of each hemisphere obtained. The coronal sections were placed flat, and an 18-gauge hypodermic needle was used to separate the cortex from the remaining brain tissue. The cortical tissue was washed once in HBSS

mRNA levels were quantified by qRT-PCR and normalized relative to *ACTB* levels under the same experimental conditions. The normalized value was then expressed relative to the normalized value from DMSO treated cells. Data are the mean (±SEM) of three independent experiments. Statistical significance was determined by two-way ANOVA (*$P < 0.05$). **(C)** Quantification of *FOS* pre-mRNA containing ("unspliced") or lacking ("spliced") intron 1 in HeLa cells stimulated as above. Top: schematic of the *FOS* gene locating the primers used for qRT-PCR and for amplification of *FOS* pre-mRNA in which intron 1 is spliced (E1f and E2r) or unspliced (I1f and E2r). Data are the mean (±SEM) of three independent experiments. Statistical significance was determined by two-way ANOVA (****$P < 0.0001$). **(D)** GFP-FUS binding at the indicated regions of the rDNA loci was quantified by chromatin immunoprecipitated/qPCR in HeLa cells following incubation with DMSO vehicle or 4 $\mu$M CPT for 45 min, with or without co-incubation with 10 $\mu$M CX5461 (Pol I inhibitor) for 3 h. Top: schematic showing the regions of the rDNA repeats amplified by qPCR are indicated. *ACTIN* was also amplified and quantified as a control. Data are the mean (±SEM) of four independent experiments. Statistically nonsignificant ("ns") and significant (two-tailed *t* test; *$P < 0.05$, **$P < 0.01$) differences are indicated. **(E)** Anti-RNA:DNA hybrid (S9.6 antibody) ChIP-qPCR at the indicated regions of the rDNA locus in HeLa cells after mock treatment (DMSO) or treatment with 4 $\mu$M CPT for 45 min with or without preincubation with 10 $\mu$M CX5461 for 3 h. Data are the mean (±SEM) of six independent experiments.

before incubation in pre-warmed HBSS containing 0.04% trypsin for 15 min (inverting the tube every 5 min) at 37°C. DNAse (D5025; Sigma-Aldrich) was added to the solution (0.06 mg/ml) before centrifuging the cells at 300 *g* for 10 min. The cell pellet was resuspended in HBSS containing 1% AlbuMAX (11020-013; Gibco), 0.5 mg/ml of trypsin inhibitor (T9003; Sigma-Aldrich), and DNase before dissociating the cells using a flame polished glass pipette until a single cell suspension was obtained. Neurons were plated at a density of 50,000 per 13-mm glass coverslip precoated for 24 h with 0.1 mg/ml of poly-D-lysine (P6407; Sigma-Aldrich) in Neurobasal media (11570556; Gibco) supplemented with 1% L-glutamine (11500626; Gibco), 1% P/S, and 1× B27 supplement (17504-044; Thermo Fisher Scientific).

## Differentiation of human neural progenitor cells (NPCs) to spinal MNs

The generation and culturing of human NPCs has been previously described (37). NPCs were grown in N2B27 medium (DMEM-F12/Neurobasal at 50:50 supplemented with 1% penicillin/streptomycin/glutamine, 1% B27 supplement without vitamin A, and 0.5% N2 supplement) containing 3 $\mu$M CHIR99021 (Cayman), 150 $\mu$M L-ascorbic acid (Sigma-Aldrich), and 0.5 $\mu$M Smoothened Agonist (SAG) (Cayman). For differentiation, the medium was replaced with N2B27 containing 1 ng/ml BDNF (Miltenyi Biotec), 0.2 mM L-ascorbic acid, 1 $\mu$M retinoic acid (Sigma-Aldrich), 1 ng/ml glial cell line-derived neurotrophic factor (GDNF) (Miltenyi Biotec), and 0.5 $\mu$M SAG. On day 8, the medium was changed for inducing neural maturation to N2B27 containing 5 ng/ml activin A (Miltenyi Biotec), 0.1 mM dbcAMP (Sigma-Aldrich), 2 ng/ml BDNF, 0.2 mM L-ascorbic acid, 1 ng/ml TGF$\beta$-3 (Peprotech), and 2 ng/ml GDNF. On day 10, the cells were seeded on a four-well plate for immunofluorescence and 2.5 $\mu$M N-[(3,5-Difluorophenyl)acetyl]-L-alanyl-2-phenyl]glycine-1,1-dimethylethyl ester (DAPT) (Sigma-Aldrich) and activin were added to the maturation medium. After 2 d, DAPT and acitivin were removed and the cells were further grown in the maturation medium for neural maturation until day 30.

## Antibodies

Primary antibodies used were mouse monoclonal anti-phospho-$\gamma$H2AX (Ser139) (05-636; 1:1,000 dilution; Merck-Millipore), rabbit polyclonal anti-53BP1 (A300-272A; 1:400; Bethyl Laboratories), rabbit polyclonal anti-CENPF (ab5; 1:500 dilution; Abcam), rabbit polyclonal anti-FUS (NB100-565; 1:400 dilution; Novus), mouse monoclonal anti-FUS (sc-47711; 1:400; Santa Cruz), mouse monoclonal anti-fibrillarin (ab4566; 1:500; Abcam), rabbit polyclonal anti-fibrillarin (ab5821; 1:500; Abcam), and mouse monoclonal anti-nucleophosmin (B23, FC-61991; 1:500; Thermo Fisher Scientific). For chromatin immunoprecipitation experiments, chromatin immunoprecipitation (ChIP)-grade anti-GFP (ab290; 5 $\mu$g/IP; Abcam) and mouse monoclonal anti RNA:DNA hybrids (S9.6, ENH001; 5 $\mu$g/IP; Kerafast) were used. As neural markers, mouse monoclonal anti-MAP2 (MAB3418; 1:1,000; Millipore) and rabbit polyclonal anti-ChAT (AB143; 1:400; Merck) were used.

## EU pulse labelling, immunofluorescence, and microscopy

Cells were grown on coverslips, treated with IR/chemicals or not as indicated above and in the text, and fixed for 10 min in 4%

paraformaldehyde in PBS, or for detection of endogenous FUS in 1% paraformaldehyde in PBS. After fixation, the cells were washed twice in PBS, permeabilized for 2 min in 0.2% Triton X-100 in PBS, blocked for 1 h in 5% BSA in PBS, and incubated with the indicated primary antibody for 2 h in 1% BSA in PBS. The cells were then washed (3 × 10 min) in 0.1% Tween-20 in PBS and incubated for 1 h with the appropriate Alexa Fluor–conjugated secondary antibody (1:1,000 dilution in 1% BSA in PBS). The cells were counterstained with DAPI (4′,6-diamidino-2-phenylindole; Sigma-Aldrich) and mounted in Vectashield (Vector Labs). For $\gamma$H2AX immunostaining, cells on coverslips were arrested by serum starvation by growth in MEM containing 0.1% FCS for 4 d and then treated or not with CPT/Etoposide/IR as described above and in the text. $\gamma$H2AX immunofoci were counted (double-blind) in 40 G1-phase cells from each experimental sample, defined by CENPF immunostaining as CENPF-negative. For pulse labelling with 5-EU, the cells were incubated with 1 mM EU (Invitrogen) for 30 min, fixed, and subjected to Click Chemistry using a Click-iT Plus Alexa Fluor 647 Picolyl Azide imaging kit (Thermo Fisher Scientific) according to the manufacturer's specifications. Mean fluorescence was quantified by Olympus ScanR analysis software from >500 cells for each experiment. High-resolution microscopy of fixed samples was carried out using either (for HeLa cell and motor neurons) a Zeiss Apotome AxioObserver Z1 epifluorescence microscopy system with 40×/1.3 oil Plan-Apochromat objective, Hamamatsu ORCA-Flash4.0 LT camera and ZEN 2 core imaging software, or (for A549 cells and fibroblasts) a Zeiss Axioplan 2 with Plan-Apochromat 100×/1.4 oil DIC lenses, MicroManager software for acquisition and ImageJ for image processing, or (for mouse cortical neurons) a Leica TSC SP8 confocal microscope with a 63× objective. Automated wide-field microscopy was performed on an Olympus ScanR system (motorized IX83 microscope) with ScanR Image Acquisition and Analysis Software, 40×/0.6 (LUCPLFLN 40× PH) dry objectives and Hamamatsu ORCA-R2 digital CCD camera C10600.

## *FOS* gene expression and splicing

The cells were serum-starved for 3 h and transcription induced by addition of A23187 for the time points detailed in the text. After 20 min, CPT was either added or not added. RNA was extracted from pelleted cells using the RNeasy kit (QIAGEN) with an additional DNase step. Total RNA (1 $\mu$g) and oligo (dT) (0.16 $\mu$g; Ambion) were heated at 70°C for 5 min, chilled on ice, and reverse transcribed for 2 h at 42°C. The cDNA was treated with RNase at 37°C for 30 min and purified using a PCR purification kit (QIAGEN). Aliquots of 2.5 $\mu$l were used in qRT-PCR (25-$\mu$l total volume). The cDNA was analyzed by quantitative PCR (qPCR) by primers in exon 4. The expression data were first normalized to the data for *ACTB* in the same experimental condition and then to DMSO-uninduced control. For studying splicing of intron 1, a primer located in *FOS* intron 2 was used for reverse transcription. The ratio spliced/unspliced was analyzed by qPCR with primers spanning exon 1–exon 2 (spliced intron 1), and intron 1–exon 2 (unspliced intron 1). Primer sets are detailed in Table S1.

## Chromatin immunoprecipitation

The cells were cross-linked with 1% formaldehyde at room temperature for 10 min, followed by the addition of glycine to 125 mM

for 5 min at room temperature to terminate cross-linking. Cells pellets were recovered by scraping, centrifugation, and resuspended in 0.3 ml lysis buffer (0.5% SDS, 5 mM EDTA, 50 mM Tris–HCl, pH 8, 1 mM DTT, 50 $\mu$g/ml PMSF, and 1× protease inhibitor cocktail [P8340; Sigma-Aldrich]). The cells were then sonicated in a Bioruptor at maximum intensity for either 10 min in 20 cycles (30 s off, 30 s on) for GFP ChIP or for 15 min in 30 cycles (30 s off, 30 s on) for S9.6 ChIP followed by clarification by centrifugation. Supernatants were collected, 1/30 volume reserved as whole-cell extract, and the remainder diluted in 1% Triton X-100, 200 mM NaCl, 10 mM Tris–HCl, pH 8, 2 mM EDTA, 50 $\mu$g/ml PMSF, and 1× protease inhibitor cocktail. Diluted extract was precleared with magnetic beads (Dynabeads; Invitrogen) that were blocked with sheared salmon sperm DNA for 1 h at 4°C. Blocked beads were incubated with 5 $\mu$g per sample of anti-GFP antibody (ChIP-grade; Abcam) or anti-RNA:DNA hybrid antibody (S9.6; Kerafast) for 1 h at room temperature. The precleared extract was then added, and immunoprecipitation was performed for 4 h at 4°C. The beads were washed in NaCl buffer (0.1% SDS, 1% Triton X-100, 2 mM EDTA, 20 mM Tris–HCl, pH 8, and 500 mM NaCl) followed by LiCl buffer (1% deoxycholate, 1% NP-40, 1 mM EDTA, 10 mM Tris–HCl, pH 8, and 250 mM LiCl). The beads were then washed two times with TE buffer, and bound material was eluted with 2% SDS in 1× TE at 65°C for 20 min. Eluates were heated at 65°C for >6 h to reverse cross-links and treated with 100 $\mu$g of proteinase K at 50°C for 1 h. DNA fragments were purified (QIAquick PCR purification kit; QIAGEN) and eluted in 100 $\mu$l of milliQ water. 3 $\mu$l of the immunoprecipitated material or whole-cell extract (1:100 dilution) was used per 12-$\mu$l reaction of a SYBR Green–based quantitative real-time PCR using a Stratagene MX3005P system. The set of primers is detailed in Table S1.

### Clonogenic survival assays

Control and patient fibroblasts were plated in 10-cm dishes and were allowed to settle for 24 h before being either treated with indicated concentrations of CPT (nM) or irradiated with indicated doses (Gy). The cells were allowed to grow for 19 d in the presence of CPT or after irradiation. The colonies were rinsed in PBS and fixed/stained in 70% ethanol/1% methylene blue. HeLa cells were plated 4 h before treatment with the indicated concentrations of CPT (nM), allowed to grow for 12 d, and stained as described earlier. The surviving fraction at each dose was calculated by dividing the average number of colonies in treated dishes by the average number in untreated dishes.

### Data statement

All data related to this manuscript is available upon request and will be deposited for public access at https://sussex.figshare.com/.

# Supplementary Information

# Acknowledgements

We thank Don Cleveland for the HeLa cell lines and Wim Robberecht for ALS fibroblasts. We also thank William Gittens, Nigel Leigh, and Hannes Glass for useful discussions. This work was funded by an MRC project grant to KW Caldecott and M Hafezparast (MR/K01854X/1), and MRC and ERC Programme grants to KW Caldecott (ERC SIDSCA 694996, MR/P010121/1). MI Martinez-Macias was also supported by an European Molecular Biology Organisation (EMBO) Long-Term Fellowship ALTF 751-2013. We also thank Hans and Marit Rausing for their generous support and Scholarship awards to RL Green/M Hafezparast and DAQ Moore/KW Caldecott. The work was additionally supported by the Helmholtz Virtual Institute (VH-VI-510) "RNA dysmetabolism in ALS and frontotemporal dementia" to A Hermann, the NOMIS foundation to A Hermann, and a family of a deceased ALS patient and the Stiftung Hochschulmedizin Dresden and the Hermann und Lilly Schilling-Stiftung für medizinische Forschung im Stifterverband to A Hermann.

## Author Contributions

MI Martinez-Macias: conceptualization, formal analysis, investigation, and writing—original draft, review, and editing.
DAQ Moore: investigation and methodology.
RL Green: investigation and methodology.
F Gomez-Herreros: methodology and writing—review and editing.
M Naumann: formal analysis, methodology, and writing—review and editing.
A Hermann: formal analysis, methodology, and writing—review and editing.
P Van Damme: methodology.
M Hafezparast: supervision.
KW Caldecott: conceptualization, formal analysis, supervision, funding acquisition, project administration, and writing—original draft, review, and editing.

## Conflict of Interest Statement

The authors declare that they have no conflict of interest.

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
