## [Reviewer comments · Life Science Alliance]

Life Science Alliance

FUS is part of the cellular response to topoisomerase-induced DNA breaks and transcriptional stress

Maria Martinez-Macias, Duncan Moore, Ryan Green, Fernando Gomez-Herreros, Marcel Naumann, andreas hermann, Philip Van Damme, Majid Hafezparast, and Keith Caldecott
DOI: <https://doi.org/10.26508/lsa.201800222>

Corresponding author(s): Keith Caldecott, University of Sussex and Maria Martinez-Macias, University of Sussex

Review Timeline:

Submission Date:	2018-10-23
Editorial Decision:	2018-12-12
Revision Received:	2019-01-28
Editorial Decision:	2019-02-11
Revision Received:	2019-02-13
Accepted:	2019-02-15

Scientific Editor: Andrea Leibfried

Transaction Report:

December 12, 2018

Re: Life Science Alliance manuscript #LSA-2018-00222-T

Prof. Keith W Caldecott
University of Sussex
Genome Damage and Stability Centre
Science Park Road
Falmer
Brighton, Sussex BN1 9RQ
United Kingdom

Dear Dr. Caldecott,

Thank you for submitting your manuscript entitled "FUS (Fused in Sarcoma) is a component of the cellular response to topoisomerase 1-induced DNA breakage and transcription stress" to Life Science Alliance. The manuscript was assessed by expert reviewers, whose comments are appended to this letter.

As you will see, the reviewers appreciate your observations. While reviewer #3 supports publication almost as is, reviewer #1 and #2 raise some concerns that should get addressed in a revision. We would thus like to invite you to submit a revised version of your work. Importantly, text and data representation need to be changed, quantifications added and the discussion should get extended (reviewer #1). This reviewer also thinks that HeLa cells are not a good choice for the analysis in Figure 7, so please discuss the limitations of this choice. Please also test another RNA synthesis inhibitor and add further discussion to address point 3 of reviewer #2. Ideally, imaging data for the mutants tested should get provided as well (reviewer #2, point 2), but we do not expect you to elucidate the mechanism driving translocation.

Thank you for this interesting contribution to Life Science Alliance. We are looking forward to receiving your revised manuscript.

Sincerely,

- A letter addressing the reviewers' comments point by point.
- An editable version of the final text (.DOC or .DOCX) is needed for copyediting (no PDFs).
- High-resolution figure, supplementary figure and video files uploaded as individual files: See our detailed guidelines for preparing your production-ready images, <http://life-science-alliance.org/authorguide>
- Summary blurb (enter in submission system): A short text summarizing in a single sentence the study (max. 200 characters including spaces). This text is used in conjunction with the titles of papers, hence should be informative and complementary to the title and running title. It should describe the context and significance of the findings for a general readership; it should be written in the present tense and refer to the work in the third person. Author names should not be mentioned.

B. MANUSCRIPT ORGANIZATION AND FORMATTING:

Full guidelines are available on our Instructions for Authors page, <http://life-science-alliance.org/authorguide>

Reviewer #1 (Comments to the Authors (Required)):

This manuscript describes Fused in sarcoma (FUS) as a component of the response to TOP1-induced DNA damage. They hypothesize FUS moves to nucleoli to either affect RNA processing or modulate rRNA synthesis. They also show that FUS-mutant human fibroblasts from ALS are hypersensitive to TOP1 breakage, connecting FUS ALS to DNA damage, which is becoming an emerging theme in many age-onset neurodegenerative diseases.

There is a problem in figure 7 with the use of HeLa cells.

The repeatedly mention a similar effect with GFP-TDP43, citing proof of relevance to ALS, but they lack any data with TDP mutants or TDP mutant ALS fibroblasts.

The discussion of rRNA synthetic downregulation in pages 8-9 is far too speculative without any mechanistic data.

Critical points:

Figure 2 could benefit from zoomed panels to better focus on the nucleoli. They have to show the DAPI channel alone, not just FUS and FUS/DAPI. (as they did for GFP-FUS and fibrillarin).

Figure 3: the signal appears blown out in both ChAT and DAPI channels in 3C, the zoomed areas should not appear overlaid over the primary image, but in adjacent panels. The figure has a problem with qualitative presentation of data, often with one cell. This could be a problem with reproduction, upon CPT treatment, how many cells reveal nucleolar localization? These observations need to be quantified and scored on at least 100 cells.

Figure 7 has a "normal" versus FUS ALS fibroblast lines, which may be problematic because the lines are not isogenic, and there is no information about these lines -are they age matched? Same sex? ALS patients tend to be older, and we know ALS has a chronic ROS load as well. The controls of "isogenic" hela cells is very problematic, as these lines are known to have shattered genomes and have all kinds of problems with genomic stability and have a very abnormal replication rate and hyperactive DNA repair. The experiment would better to have been done in RPE1 cells expressing GFP-FUS moieties. This experiment concludes a dominant phenotype of FUS mutants, but this mechanism was not discussed.

FET proteins are known to be liquid-liquid phase separated, a process known to regulated by PAR, yet they see no effects of PARP1 inhibition on the movement to nucleoli - is it possible FUS relocation is just along with rRNA relocalization? A potential experiment could be to see if RNA in nucleoli are gelated during stress, which could be shown by lack of FRAP recovery of GFP-FUS, and could explain inhibition of rRNA.

Jain and Vale, Nature. 2017 Jun 8; 546(7657): 243-247.

Reviewer #2 (Comments to the Authors (Required)):

Caldecott and coworkers demonstrate that FUS accumulates in nucleoli and rDNA in response to the topoisomerase I inhibitor camptothecin, while not affecting the chromatin response measured

by histone gamma-H2AX and conferring mild resistance to camptothecin. The authors also show that actinomycin at micromolar concentration induces the accumulation/translocation of FUS to nucleoli. The data are clearly presented and convincing. The references are informative, and the text is concise and logical. Yet, the study remains descriptive.

Our comments and suggestions are listed below:

1. The mechanistic implications of the novel finding that FUS concentrates in nucleoli in response to camptothecin remain obscure, and the manuscript does not provide mechanism for the translocation. The only insight is that actinomycin acts like camptothecin. Yet, actinomycin D has been shown to also act as a topoisomerase I inhibitor (for instance: Trask DK & Muller MT 1988 PNAS; Wassermann K et al 1990 Mol Pharmacol). Thus, the authors should test other RNA synthesis inhibitors such as DRB (see Buckwalter CA et al 1996 Cancer Res).
2. To establish the relationship between survival and nucleolar translocation, the authors should consider testing the ALS mutants tested in Fig. 7 by microscopy (like in Figs. 1-5). Are those mutants defective in translocation? What drives the translocation?
3. The discussion and introduction should mention and discuss the reported fact that topoisomerase I is rapidly excluded/translocated from nucleoli in response to camptothecin and RNA synthesis inhibitors (see Buckwalter CA et al 1996 Cancer Res). Are the recruitment of FUS to nucleoli related to the exclusion of topoisomerase I from nucleoli?
4. Other editorial points:
 - a. Please use roman numeral "I" for the full name of topoisomerase I (instead of "1") (Line 2 and throughout the manuscript). The abbreviation TOP1 is OK.
 - b. Line 262: please remove "res" or replace with "refs".
 - c. Line 654: the figure shows "murine" cells (?)

Reviewer #3 (Comments to the Authors (Required)):

This paper reports that FUS, commonly mutated in ALS, translocates to the nucleolus when cells are exposed to the TOP1 inhibitor camptothecin (CPT). Radiation which induces DSBs does not cause nucleolar translocation. Data is presented in favour of a model suggesting that it is perturbation of POLII transcription by CPT that causes translocation to the nucleolus, which also requires ongoing POLI transcription in the nucleolus.

The paper is straightforward and the data are of a high technical quality. The conclusions are justified by the data. The speculative explanations for the underlying mechanisms are reasonable. Ideally at least one other TOP1 inhibitors besides CPT would have been included but this is not strictly necessary for acceptance of the paper.

Response to the Editor and Referees

Editor: Text and data representation need to be changed, quantifications added and the discussion should get extended (reviewer #1). This reviewer also thinks that HeLa cells are not a good choice for the analysis in Figure 7, so please discuss the limitations of this choice. Please also test another RNA synthesis inhibitor and add further discussion to address point 3 of reviewer #2. Ideally, imaging data for the mutants tested should get provided as well (reviewer #2, point 2), but we do not expect you to elucidate the mechanism driving translocation.

Thank you for this considered response. We have addressed all of these comments, and those of the referees, below.

Best wishes (and thank you to the referees)

Keith

Reviewer #1: The repeatedly mention a similar effect with GFP-TDP43, citing proof of relevance to ALS, but they lack any data with TDP mutants or TDP mutant ALS fibroblasts. We have now quantified the response of both GFP-FUS and GFP-TDP-43 in response to TOP1-induced DNA breakage, and added these data to the figures (please note that the TDP-43 data has been moved to Fig.S1). We have also added data for both FUS and TDP-43 mutations associated with ALS. The latter data indicate that ALS-associated mutations do not markedly alter the movement of FUS or TDP-43 into nucleoli, arguing that a defect simply in the movement of these proteins to nucleoli cannot account for the disease. Of course, it is possible that the functionality of these proteins within nucleoli is defective. We address these points in the text (Page 5 first paragraph and Page 10, second paragraph).

The discussion of rRNA synthetic downregulation in pages 8-9 is far too speculative without any mechanistic data. We have now shortened the discussion of this possibility (Page 10).

Figure 2 could benefit from zoomed panels to better focus on the nucleoli. They have to show the DAPI channel alone, not just FUS and FUS/DAPI. (as they did for GFP-FUS and fibrillarin). Figure 2 has now been modified as requested and the quantification added to the figure.

Figure 3: the signal appears blown out in both ChAT and DAPI channels in 3C, the zoomed areas should not appear overlaid over the primary image, but in adjacent panels. The figure has a problem with qualitative presentation of data, often with one cell. This could be a problem with reproduction, upon CPT treatment, how many cells reveal nucleolar localization? These observations need to be quantified and scored on at least 100 cells. Fig.3 is modified as requested and quantification added.

Figure 7 has a "normal" versus FUS ALS fibroblast lines, which may be problematic because the lines are not isogenic, and there is no information about these lines -are they age matched? Same sex? ALS patients tend to be older, and we know ALS has a chronic ROS load as well. A description of the fibroblast lines has been added (Page 11). The 'normal' fibroblasts are from a clinically unaffected 44-yr old male. Whilst not isogenic, the affected ALS patient is a 33-yr old female harbouring the FUS mutation R521H and is a sibling of the unaffected male.

The controls of "isogenic" hela cells is very problematic, as these lines are known to have shattered genomes and have all kinds of problems with genomic stability and have a very abnormal replication rate and hyperactive DNA repair. The experiment would better to have been done in RPE1 cells expressing GFP-FUS moieties. This experiment concludes a dominant phenotype of FUS mutants, but this mechanism was not discussed. We agree that cancer cell lines such as HeLa have caveats in

terms of genetic stability, but the advantage of the 'isogenic' HeLa cell lines employed here is that the tagged alleles are expressed at physiologically-relevant levels and have been robustly characterized in the Cleveland lab (references 35 and 36 in the text). We believe that the use of these cells, in combination with primary fibroblasts from an ALS patient, primary mouse cortical neurones, and iPSC-derived human motor neurones, provide convincing evidence that the phenomenon we describe here for FUS re-localisation is physiologically relevant.

FET proteins are known to be liquid-liquid phase separated, a process known to regulated by PAR, yet they see no effects of PARP1 inhibition on the movement to nucleoli - is it possible FUS relocation is just along with rRNA relocalization? Indeed. It is possible that the relocalisation of FUS is promoted by an association with rRNA, either directly or indirectly. We have now included this point in our discussion (Page 9, penultimate paragraph)

A potential experiment could be to see if RNA in nucleoli are gelled during stress, which could be shown by lack of FRAP recovery of GFP-FUS, and could explain inhibition of rRNA. Jain and Vale, Nature. 2017 Jun 8; 546(7657): 243-247. We agree that this is an interesting idea and experiment, but feel it is beyond the scope of the current paper.

Reviewer #2:

1. The mechanistic implications of the novel finding that FUS concentrates in nucleoli in response to camptothecin remain obscure, and the manuscript does not provide mechanism for the translocation. The only insight is that actinomycin acts like camptothecin. Yet, actinomycin D has been shown to also act as a topoisomerase I inhibitor (for instance: Trask DK & Muller MT 1988 PNAS; Wassermann K et al 1990 Mol Pharmacol). Thus, the authors should test other RNA synthesis inhibitors such as DRB (see Buckwalter CA et al 1996 Cancer Res). We have now tested DRB as requested. This treatment induced accumulation of wild type and mutant FUS in nucleoli, strengthening our conclusions (Fig.5D & Fig.S3).

2. To establish the relationship between survival and nucleolar translocation, the authors should consider testing the ALS mutants tested in Fig. 7 by microscopy (like in Figs. 1-5). Are those mutants defective in translocation? What drives the translocation? We have now included data for both mutant FUS and TDP-43. The ALS-associated mutations do not markedly affect the nucleolar localization(Fig.7 & Fig.S1A). We thus suggest that (Page 10, penultimate paragraph), "whilst one or more of the roles played by FUS following relocalisation may be defective in ALS, the relocalisation process itself is not defective, because this phenomenon was similarly observed by FUS proteins that harbour ALS-associated mutations".

3. The discussion and introduction should mention and discuss the reported fact that topoisomerase I is rapidly excluded/translocated from nucleoli in response to camptothecin and RNA synthesis inhibitors (see Buckwalter CA et al 1996 Cancer Res). Are the recruitment of FUS to nucleoli related to the exclusion of topoisomerase I from nucleoli? We thank referee for highlighting this interesting point and we have now discussed it in the text (Page 9, 1st paragraph).

4. a. Please use roman numeral "I" for the full name of topoisomerase I (instead of "1") (Line 2 and throughout the manuscript). The abbreviation TOP1 is OK. Done, thank you.

b. Line 262: please remove "res" or replace with "refs". Done

c. Line 654: the figure shows "murine" cells (?) We have now corrected any discrepancies - these are HeLa cells. Note that the TDP-43 data has now been moved to Fig.S1, to make room for other requested edits.

Reviewer #3: No changes required, thank you.

February 11, 2019

RE: Life Science Alliance Manuscript #LSA-2018-00222-TR

Prof. Keith W Caldecott
University of Sussex
Genome Damage and Stability Centre
Science Park Road
Falmer
Brighton, Sussex BN1 9RQ
United Kingdom

Dear Dr. Caldecott,

Thank you for submitting your revised manuscript entitled "FUS is part of the cellular response to topoisomerase-induced DNA breaks and transcriptional stress". As you will see, the reviewers appreciate the introduced changes and we would thus be happy to publish your paper in Life Science Alliance pending final revisions necessary to meet our formatting guidelines:

- please address the comment made by reviewer #1 regarding callouts
- please list 10 authors et al. in the reference list
- please check and add scale bars where missing

A. FINAL FILES:

-- High-resolution figure, supplementary figure and video files uploaded as individual files: See our detailed guidelines for preparing your production-ready images, <http://life-science-alliance.org/authorguide>

-- Summary blurb (enter in submission system): A short text summarizing in a single sentence the study (max. 200 characters including spaces). This text is used in conjunction with the titles of papers, hence should be informative and complementary to the title. It should describe the context

and significance of the findings for a general readership; it should be written in the present tense and refer to the work in the third person. Author names should not be mentioned.

B. MANUSCRIPT ORGANIZATION AND FORMATTING:

Full guidelines are available on our Instructions for Authors page, <http://life-science-alliance.org/authorguide>

Sincerely,

Reviewer #1 (Comments to the Authors (Required)):

The authors have addressed most of my concerns. I still do not agree the HeLa lines are a proper model, and they certainly do not have physiological stoichiometry as they are all transgenic.

They should remove the use of an ampersand in place of "and" in Figure callouts.

Reviewer #2 (Comments to the Authors (Required)):

The revisions are acceptable. The manuscript is now publishable.

February 15, 2019

RE: Life Science Alliance Manuscript #LSA-2018-00222-TRR

Prof. Keith W Caldecott
University of Sussex
Genome Damage and Stability Centre
Science Park Road
Falmer
Brighton, Sussex BN1 9RQ
United Kingdom

Dear Dr. Caldecott,

Thank you for submitting your Research Article entitled "FUS is part of the cellular response to topoisomerase-induced DNA breaks and transcriptional stress". It is a pleasure to let you know that your manuscript is now accepted for publication in Life Science Alliance. Congratulations on this interesting work.

DISTRIBUTION OF MATERIALS:

Again, congratulations on a very nice paper. I hope you found the review process to be constructive and are pleased with how the manuscript was handled editorially. We look forward to future exciting

submissions from your lab.

Sincerely,
